# Exploring breast cancer awareness and screening practices amongst rural women in The Gambia: Community-based cross-sectional study

**Bakary Kinteh**[1,2]*, **Fatoumatta Jitteh**[2,3], **Mansour Badjie**[1], **Amadou Barrow**[1,4,5], **Lamin Jaiteh**[2,3]

**1** School of Public Health, Gambia College, Brikama, The Gambia, **2** Solace Foundation, Banjul, The Gambia, **3** Edward Francis Small Teaching Hospital, Banjul, The Gambia, **4** Department of Public & Environmental Health, School of Medicine & Allied Health Sciences, University of The Gambia, Serrekunda, The Gambia, **5** Department of Epidemiology, University of Florida, Gainesville, Florida, United States of America

* bakarykinteh461@gmail.com

## Abstract

### Background

Breast cancer is a significant public health challenge in The Gambia, where it ranks as the second most commonly diagnosed cancer among women. Rural women face unique challenges in accessing screening services; however, evidence about their breast cancer awareness and screening uptake remains limited. This study aimed to assess breast cancer awareness and identify associated factors influencing screening uptake among rural women in The Gambia.

### Methods

A community-based cross-sectional study was conducted among rural women in The Gambia. Using multistage sampling, we recruited 985 women from two local government authorities (response rate: 90.2%). Data were collected using a structured questionnaire administered in Kobo Toolbox. Descriptive statistics were used to summarize participant characteristics, while associations between predictors and breast cancer screening uptake were assessed using Chi-squared or Fisher's exact test. Multivariable logistic regression was used to identify factors associated with screening uptake. Statistical significance was set at $p<0.05$ and adjusted odds ratios (aOR) with 95% confidence intervals were reported.

### Results

The mean age of the study population was 32 years (standard deviation [SD]: ±12.6), with 34% aged 18–24 years. Although breast cancer awareness was high (87.7%), screening uptake was low (12.6%), primarily due to limited knowledge (58.7%), service

**Data availability statement:** The anonymized data used in this study is protected under the School of Public Health, Gambia College, in accordance with its data protection and privacy framework, for which ethical approval was obtained as stated in the article. Due to these ethical restrictions, the data cannot be made publicly available. However, researchers seeking access to the data may submit requests to Solomon P.S. Jatta, the designated institutional representative for external communications, at spsjatta@gambiacollege.edu.gm. Solomon P.S. Jatta was not involved in this study and is not listed as an author. The institution ensures the long-term preservation and security of the data through a structured data protection and privacy system, with copies maintained in two independent locations to safeguard its integrity and accessibility.

**Funding:** The author(s) received no specific funding for this work.

**Competing interests:** The authors declare and affirm that they do not have any personal, financial, or professional interests that could potentially influence or tamper with the validity of the results. This work is therefore free of biases that could compromise its scientific integrity.

unavailability (13.5%) and financial constraints (13.1%). Clinical breast examination was the most common screening method used (62.6%). Multivariable analysis revealed that Students (aOR=3.111, 95% CI: 1.453–6.663) and civil servants (aOR=2.778, 95% CI: 1.174–6.573) were more likely to undergo screening compared to unemployed women. Conversely, women with two (aOR=0.061, 95% CI: 0.005–0.791), three (aOR=0.075, 95% CI: 0.006–0.967), and five children (aOR=0.065, 95% CI: 0.005–0.877) were less likely to participate in screening compared to nulliparous women.

## Conclusion

Despite the high awareness of breast cancer, screening uptake among rural women in The Gambia was notably low, primarily due to limited knowledge, service unavailability and financial constraints.

There is an urgent need for targeted interventions to improve screening uptake, particularly among multiparous women in rural communities.

## Introduction

Breast cancer remains one of the most commonly diagnosed cancers among women and is a leading cause of cancer-related deaths in women, accounting for 23% of cancer cases and 14% of deaths worldwide [1]. The high morbidity and mortality rates do not exempt low and middle countries, where the incidence of breast cancer has been increasing rapidly, largely attributed to lifestyle changes, alterations in reproductive health factors, and longer life expectancy [2]. According to recent World Health Organization (WHO) reports, 157 of 185 countries registered breast cancer as the most prevalent cancer affecting women in 2022 [3].

In The Gambia, breast cancer is the second most commonly diagnosed cancer, second only to cervical cancers with an estimated prevalence of 11.25 per100, 000 women [4]. Recent data indicates that more than 50% of diagnosed patients died of the disease by 2020 [5]. The Cancer Registry Unit under the Ministry of Health was tasked with the country's cancer data from diagnosis to treatment, faces significant challenges in meeting WHO's initiative to reduce breast cancer mortality by 3 million lives by 2024 [3]. The current limitations in capacity include breast cancer diagnosis, treatment, and patient management, had significantly impacted breast cancer outcomes. Only 50% (52/102) of healthcare facilities can provide clinical diagnosis, and women often travel up to 45 km to access healthcare facilities (Sanyang et al. 2021). Evidence suggests a 3–6 month average delay between symptom onset and seeking healthcare [6], significantly affecting prognosis and survival rates. Multiple factors contribute to poor outcomes, including limited awareness, cultural beliefs, socio-economic barriers, and inadequate access to modern diagnostic facilities [7,8].

While a recent study among female university students showed relatively good breast cancer awareness, it revealed poor screening practices, with 75% never having undergone screening [9]. The study revealed that female university students had good knowledge about breast cancer; however, breast cancer screening uptakes

were still poor, as up to 75% of the respondents had never undergone any form of breast cancer screening [9]. However, these findings may not represent rural women who face unique barriers to healthcare access and different socio-cultural contexts. Understanding breast cancer awareness and screening uptakes specifically among rural women is crucial for achieving WHO's target of reducing mortality by 25% by 2030 and 40% by 2040 through the Global Breast Cancer Initiative (GBCI) strategies of early detection, timely diagnosis, and prompt treatment [3].

The Gambian Health Policy 2021–2023 acknowledges breast cancer as a major health concern, but implementation strategies remain limited [5]. This study therefore aimed to explore breast cancer awareness and screening uptake among rural Gambian women, identify associated factors influencing uptake, and inform targeted interventions and policy decisions for improving breast cancer outcomes in underserved rural communities.

## Materials and methods

### Study design and setting

We conducted a community-based cross-sectional study between January and February 2024 to assess breast cancer awareness and screening uptake among rural women residing in the two Local Government Authorities (LGAs) in the northern regions of The Gambia. The study area comprises: Kerewan Area Council with four districts (Upper Baddibu, Lower Baddibu, Central Baddibu, and Jokadu) and Kuntaur Area Council with three districts (Upper Saloum, Lower Saloum, and Niani), with populations of 225,516 and 98,966 inhabitants, respectively [10]. Each district contains an average of 20–30 communities. Healthcare delivery in these regions is structured hierarchically, with one General Hospital in Farafenni, District hospitals in Essau, Kerewan, and Kuntaur and primary healthcare facilities consisting of six major health centers and twenty-eight community health posts [5].

The average distance to the nearest health facility offering clinical breast examination services is 15–20 kilometers, with some communities located up to 40 kilometers away [11,12]. Public transportation is irregular, and road conditions are poor, particularly during the rainy season [13]. Most women must travel 2–3 hours to access these services, incurring significant transportation costs (approximately 200–300 Dalasi per trip, equivalent to 3–5 USD) [14]. Additionally, clinical breast examination services, while subsidized, still cost about 100 Dalasi (approximately 1.5 USD), representing a significant financial burden for many rural families. The nearest mammography services are available only at the Edward Francis Small Teaching Hospital in Banjul [12], requiring women to travel over 200 kilometers from most study communities [13]. This distance, combined with the high cost of mammography (approximately 2000 Dalasi or 30 USD), makes regular screening particularly challenging for rural women.

### Study population

The study population comprised women aged ≥ 18 years residing in rural communities within two local government authorities in Northern Gambia. The eligibility criteria included a minimum residential duration of 12 months in the study communities prior to data collection, ensuring that the participants had adequate exposure to local healthcare services and community practices.

### Study variables

The primary outcome variable was breast cancer screening uptake, measured as a binary variable (yes/no) based on whether the respondents had ever undergone any form of breast cancer screening (breast self-examination, clinical breast examination, or mammography) at least once in their lifetime. Independent variables included sociodemographic characteristics: age (categorized as 18–24, 25–31, 32–38, 39–45, and 46 years & above); ethnicity (Mandinka, Fula, Wolof, Serahuli, Jola, Serer, and others); religion (Islam, Christianity); nationality (Gambian, non-Gambian); educational level (no formal education, primary, secondary, and tertiary); current occupation (unemployed, civil servant, student, housewife,

farmer, business, and others); marital status (single, married, divorced/separated, and widowed); and parity (nullipara, primipara, two, three, four, five, and six & above).

## Sample size determination

The sample size was calculated using the Epi Info version 7.0 (Centers for Disease Control and Prevention, Atlanta, GA, USA). Assuming a 50% prevalence of breast cancer awareness (as no previous community-based studies were available), 99% confidence level, and precision of 5%, the minimum required sample size was determined using the following formula:

$$n = \left[ Z^2_{1-a/2} \times P(1-P) \right] / d^2$$

where $n$ is the sample size, $Z_{1-a}/_2$ is the standard normal variate at 99% confidence level (2.576), $P$ is the expected proportion (50%), and $d$ is the precision (5%). The calculated minimum sample size was 662 respondents. After accounting for a 10% non-response rate and a design effect of 1.5, owing to the multi-stage sampling technique, the final target sample size was 1,092. Of these, 985 completed the study, yielding a response rate of 90.2%.

## Sampling strategy

A four-stage sampling method was employed. First, the northern region was purposively selected due to its geographical location, high population density of underserved rural women, and absence of mammography services. This selection aligns with the study's aim of understanding screening uptake in rural, as it represents a significant proportion of rural communities with limited healthcare access. The northern region of The Gambia comprises of Kerewan and Kuntaur local government areas (LGAs), with (Upper Baddibu, Lower Baddibu, Central Baddibu, and Jokadu) and three (Upper Saloum, Lower Saloum, and Niani) districts, respectively. Second, five communities were randomly selected from each district, yielding 35 study communities in total. Third, within each selected community, households were sampled using probability proportional to size, selecting an average of 25–35 households per community. Finally, one eligible woman was selected from each household. In households with multiple eligible respondents, a participant was selected using a random number generator to ensure unbiased selection and minimize selection bias that could arise from choosing the most available or willing participant. When no eligible respondents were found in a selected household, it was documented as a non-eligible household in our response rate calculations, and data collectors proceeded to the next household following a predetermined clockwise direction until the required sample size was achieved.

## Data collection instruments

Data were collected using a structured questionnaire adapted from a previous study of female university students in The Gambia (9), modified for community-level applications. The questionnaire was comprised of three main sections: sociodemographic characteristics, breast cancer awareness, and screening practices. Sociodemographic variables included age (in years), ethnicity, religion, nationality, educational level, current occupation, marital status, and parity. Breast cancer screening practices were assessed in four domains: screening history, screening method (breast self-examination, clinical breast examination, or mammography), facilitating factors for screening uptake, and barriers to screening.

The questionnaire was digitized using the Kobo Toolbox platform for electronic data capture. Data collectors were recruited from the University of The Gambia School of Medicine and Allied Health Sciences and the School of Public Health at Gambia College by the Solace Foundation. All data collectors underwent a one-day standardized training on the Kobo Toolbox application, interviewing techniques, and translation protocols for three major local languages (Mandinka, Wolof, and Fula), which collectively cover approximately 92% of the study area's population. To ensure instrument

reliability and consistency, the questionnaire was pilot tested among women in the West Coast Region prior to field implementation.

Field data collection was conducted from January 24–31, 2024, by 150 trained research assistants. Quality control measures included field supervision and real-time data monitoring using the Kobo Toolbox server. The dataset was subsequently exported to Microsoft Excel for data cleaning and validation. This systematic approach to data collection and management helped to ensure data quality and completeness throughout the study period.

## Data management and statistical analysis

Data analysis was conducted using IBM SPSS Statistics version 28. Descriptive statistics such as means with standard deviations (SD) for continuous variables, and frequencies and percentages for categorical variables were used to present the women characteristics including their breast cancer awareness and screening practices. As appropriate, Pearson's chi-squared or Fisher's exact test was used to examine the associations between breast cancer screening uptake and potential covariates.

Prior to modeling, we assessed multicollinearity among predictor variables using variance inflation factors (VIF), where a VIF >10 was considered indicative of high multicollinearity. The educational level showed high collinearity with occupational status and was subsequently removed to prevent model overfitting. Despite its non-significance at the bivariate analysis level, ethnicity was retained in the model owing to its theoretical importance and our specific interest in understanding its effects on breast cancer screening uptake in relation to other predictors.

Variables with p-values <0.20 in the bivariate analysis were included in the multivariable model. This less conservative threshold was chosen to minimize Type II errors and retain potentially important predictors that might not meet the conventional p<0.05 criterion at the bivariate level. Despite its non-significance at the bivariate analysis level, ethnicity was retained in the model owing to its theoretical importance and our specific interest in understanding its effects on breast cancer screening uptake in relation to other predictors. We employed multivariable logistic regression owing to the binary nature of our outcome variable (breast cancer screening uptake: yes/no) and its suitability for modeling multiple predictors simultaneously while controlling for potential confounders. A backward stepwise selection method was used, starting with all eligible variables and iteratively removing the least significant predictors while retaining the variables of theoretical importance regardless of statistical significance.

To identify the predictors of breast cancer screening uptake, we fitted a multivariable logistic regression model in which the outcome variable $Y$ was coded as $Y=1$ (screened) or $Y=0$ (not screened). The logistic model can be expressed as follows:

$$logit(p) = ln(p/1-p) = \beta_0 + \beta_1 X_1 + \beta_2 X_2 + ... + \beta_k X_k$$

where $p$ represents the probability of breast cancer screening uptake, $\beta_0$ is the intercept, and $\beta_1$ through $\beta_k$ are the regression coefficients for the k independent variables $(X_1...X_k)$. The odds ratio (OR) for breast cancer screening uptake can be derived as:

$$OR = exp(\beta_0 + \beta_1 X_1 + \beta_2 X_2 + ... + \beta_k X_k)$$

where $exp(\beta_i)$ represents the adjusted odds ratio (aOR) for the $ith$ predictor variable $(X_i)$, controlling for other variables in the model. Model estimates were reported as adjusted odds ratios (aORs) with 95% confidence intervals (CI). Statistical significance was declared at p<0.05.

## Ethical considerations

Ethical approval for this study was obtained from the Research Ethics Committee (REC) of Edward Francis Small Teaching Hospital (EFSTH_REC_202–038). Permission to conduct the study within the communities was verbally obtained

from community entry points, such as the Alkalos and Village Development Committees. Voluntary Informed consent was obtained from each study participant verbally or through a signed consent form, where participants could read or write before the onset of the interview and maintained throughout the data collection process.

## Results

### Women's sociodemographic characteristics

As shown in Table 1, the mean age of respondents was 32 years (SD: ±12.6), with the largest proportion (34.0%) aged between 18–24 years. The sample was predominantly Muslim (99.5%) and Gambian (97.7%). Regarding ethnicity, Mandinka (39.0%) and Wolof (37.9%) were the most common groups, followed by Fula (15.3%). Educational attainment was generally low, with nearly half (46.2%) reporting no formal education, whereas only 11.0% had attained tertiary education. Occupationally, housewives constituted the largest group (30.1%), followed by farmers (19.9%), and business owners (15.4%). Most respondents were married (71.7%), 26.8% were nulliparous, and 16.8% had six or more children.

Fig 1 presents the distribution of breast cancer screening uptake across sociodemographic characteristics. Overall, only 12.6% (n = 124) of respondents reported having ever undergone breast cancer screening, while 87.4% (n = 861) had never been screened.

### Association between the selected women's characteristics and breast cancer screening uptake

The Table 1 presents the association between the selected women's characteristics and breast cancer screening. Chi-square analysis revealed significant associations between breast cancer screening uptake and education level, current occupation, and parity. Specifically, women with tertiary education reported the highest screening rates (23.1%), compared to those with no formal education (11.0%), primary education (11.6%), and secondary education (11.6%), $\chi^2(3, N = 985) = 12.46$, $p = .006$. Among occupational groups, students (25.0%) and civil servants (22.2%) demonstrated higher screening rates than housewives (10.1%), farmers (9.2%), and business owners (11.8%), $\chi^2(5, N = 985) = 26.78$, $p < .001$. Screening uptake was inversely associated with parity, as nulliparous women (17.0%) reported the highest screening rates compared to multiparous women with six or more children (10.3%), $\chi^2(6, N = 985) = 22.64$, $p < .001$. Other variables, including age, ethnicity, religion, nationality, and marital status, were not significantly associated with screening uptake. For example, screening rates among age groups ranged from 9.5% (25–31 years) to 15.5% (18–24 years), but this was not statistically significant, $\chi^2(4, N = 985) = 5.34$, $p = .235$. Similarly, ethnicity did not exhibit significant differences, with screening rates ranging from 0% (Other ethnicities) to 25.0% (Serahuli), $\chi^2(6, N = 985) = 8.71$, $p = .463$.

### Breast cancer awareness

Table 2 below assessed breast cancer awareness among the respondents; the majority (87.7%) had heard about breast cancer prior to the study, and less than half (41.3%) reported mass media as their source of information. The majority of respondents identified obesity/overweight (35.2%), being a woman (35.0%), and having a family history of breast cancer (29.0%) as common risk factors for breast cancer. However, less than 14.5% of the respondents knew about clinical breast examinations and mammography (3.6%) as a means of early detection and diagnosis of breast cancer. Based on breast cancer prevention, the respondents identified early initiation of breastfeeding, physical exercise, and limited alcohol intake (38.8%, 37.8%, and 23.8%, respectively) as measures to prevent breast cancer among females. Nearly three-quarters (73.8%) were convinced that women of reproductive age should be screened for breast cancer.

### Breast cancer screening practices

Among those screened, clinical breast examination was the predominant method (62.6%), followed by breast self-examination (39.5%), whereas mammography utilization was minimal (1.1%) as shown in Table 3. Screening-seeking

**Table 1. Distribution of women's socio-demographic characteristics by breast cancer screening uptake.**

| Variable | Breast cancer screening uptake | | | p-valve |
|---|---|---|---|---|
| | n (%) | No (%) | Yes (%) | |
| **Age of respondents** | | | | 0.235 |
| 18–24 | 335 (34.0) | 283 (84.5) | 52 (15.5) | |
| 25–31 | 263 (26.7) | 238 (90.5) | 25 (9.5) | |
| 32–38 | 145 (14.7) | 125 (86.2) | 20 (13.8) | |
| 39–45 | 102 (10.4) | 91 (89.2) | 11 (10.8) | |
| 46 & above | 140 (14.2) | 124 (88.6) | 16 (11.4) | |
| **Ethnicity** | | | | 0.463[a] |
| Mandinka | 384 (39.0) | 345 (89.8) | 39 (10.2) | |
| Fula | 153 (15.3) | 129 (84.3) | 24 (15.7) | |
| Wolof | 373 (37.9) | 323 (86.6) | 50 (13.4) | |
| Serahuli | 8 (0.8) | 6 (75.0) | 2 (25.0) | |
| Jola | 44 (4.5) | 37 (84.1) | 7 (15.9) | |
| Serer | 21 (2.1) | 19 (90.5) | 2 (9.5) | |
| Others | 2 (0.2) | 2 (100.0) | 0 (0.0) | |
| **Religion** | | | | 0.821[a] |
| Islam | 980 (99.5) | 857 (87.4) | 123 (12.6) | |
| Christianity | 5 (0.5) | 4 (80.0) | 1 (20.0) | |
| **Nationality** | | | | 0.423[a] |
| Gambian | 962 (100.0) | 841 (87.4) | 121 (12.6) | |
| Non-Gambian | 23 (100.0) | 20 (87.0) | 3 (13.0) | |
| **Education level** | | | | **0.006**[*] |
| No formal education | 455 (46.2) | 405 (89.0) | 50 (11.0) | |
| Primary education | 164 (16.7) | 145 (88.4) | 19 (11.6) | |
| Secondary education | 258 (26.2) | 228 (88.4) | 30 (11.6) | |
| Tertiary education | 108 (11.0) | 83 (76.9) | 25 (23.1) | |
| **Current Occupation** | | | | **<0.001**[*a] |
| Unemployed | 123 (12.5) | 111 (90.2) | 12 (9.8) | |
| Civil servant | 63 (6.4) | 49 (77.8) | 14 (22.2) | |
| Student | 128 (13.0) | 96 (75.0) | 32 (25.0) | |
| Housewife | 296 (30.1) | 266 (89.9) | 30 (10.1) | |
| Farmer | 196 (19.9) | 178 (90.8) | 18 (9.2) | |
| Business | 152 (15.4) | 134 (88.2) | 18 (11.8) | |
| Others | 27 (2.7) | 27 (100.0) | 0 (0.0) | |
| **Marital status** | | | | 0.870[a] |
| Single | 218 (22.1) | 188 (86.2) | 30 (13.8) | |
| Married | 706 (71.7) | 621 (87.9) | 85 (12.1) | |
| Divorced/Separated | 22 (2.3) | 18 (81.8) | 4 (18.2) | |
| Widowed | 38 (3.9) | 33 (86.8) | 5 (13.2) | |
| **Parity** | | | | **<0.001**[*] |
| Nullipara | 264 (26.8) | 219 (83.0) | 45 (17.0) | |
| Primipara | 144 (14.6) | 134 (93.1) | 10 (6.9) | |
| Two | 125 (12.7) | 115 (92.0) | 10 (8.0) | |
| Three | 89 (9.0) | 71 (79.8) | 18 (20.2) | |
| Four | 104 (10.6) | 97 (93.3) | 7 (6.7) | |
| Five | 91 (9.2) | 76 (83.5) | 15 (16.5) | |

*(Continued)*

**Table 1.** (Continued)

| Variable | Breast cancer screening uptake | | | p-valve |
|---|---|---|---|---|
| | n (%) | No (%) | Yes (%) | |
| Six & above (Multipara) | 165 (16.8) | 148 (89.7) | 17 (10.3) | |

*Statistical significance at p-value <0.05; ª=Fisher's exact test*

behavior was universally triggered by breast pain (100%), with additional motivation from healthcare worker recommendations (58.9%) and self-detected breast lumps (40.3%). Among the non-screened participants (n=861), the primary barriers to screening were lack of awareness about screening methods and services (58.7%), followed by service unavailability (13.5%), high associated costs (13.1%), forgetfulness (11.5%), and fear of finding a mass (11.3%).

Fig 2 shows that the majority of respondents were aware that swelling of all parts of the breast (48.2%), nipple pain (45.2%), and lumps (37.1%) were common signs and symptoms of breast cancer.

### Predictors of breast cancer screening uptake

Table 4 presents the results of multivariable logistic regression analysis examining factors associated with breast cancer screening uptake. After adjusting for potential confounders, occupation and parity emerged as significant predictors of screening uptake, whereas ethnicity showed varying but non-significant associations. In terms of occupation, students demonstrated the strongest positive association with screening uptake (aOR=3.111, 95% CI: 1.453–6.663, p=0.003), being three times more likely to undergo screening compared to unemployed women. Similarly, civil servants showed significantly higher odds of screening (aOR=2.778, 95% CI: 1.174–6.573, p=0.020) compared to the unemployed group. Other occupational categories, including housewives (aOR=1.022, 95% CI: 0.489–2.135), farmers (aOR=0.936, 95% CI: 0.413–2.122), and business owners (aOR=1.138, 95% CI: 0.506–2.560), showed no significant association with screening uptake. Regarding parity, women with two children (aOR=0.061, 95% CI: 0.005–0.791, p=0.032), three children (aOR=0.075, 95% CI: 0.006–0.967, p=0.047), and five children (aOR=0.065, 95% CI: 0.005–0.877, p=0.040) were significantly less likely to undergo screening compared to nulliparous women. This inverse relationship between parity and screening uptake suggests that women with children may face unique barriers in accessing screening services.

Although not statistically significant, ethnic variations were observed, with Serer women showing nearly three times higher odds of screening (aOR=2.821, 95% CI: 0.521–15.267, p=0.229) compared to Mandinka women. Other ethnic groups, including Fula (aOR=1.552, 95% CI: 0.872–2.761) and Wolof (aOR=1.559, 95% CI: 0.977–2.486), showed modest but non-significant odds of participation in screening.

### Discussion

To attain the target set by the WHO through the Global Breast Cancer Initiative (GBCI) in The Gambia, there is a need to target all three 3 priority areas to reduce both morbidity and mortality from breast cancer. Our study examined breast cancer awareness and current screening uptake among rural women in The Gambia. The findings revealed that nearly 9 out of ten rural women knew the term breast cancer before the study. This finding is similar to a study conducted by Kinteh et al. (2023) among female University Students, where all respondents heard of breast cancer before the study, and a study conducted in Gaza, where more than 80% of respondents also heard about breast cancer [15]. The findings on breast cancer awareness were three times higher than those of a study conducted in a rural South African setting where only 31% of respondents were aware of breast cancer [16] and a study conducted in Ethiopia, where almost half (49.87%) had never heard of self-breast-examinations [17].

Mass media serves as an efficient and effective means of disseminating information to people, and they are instrumental in health promotion and education. More than 40% of our respondents had heard of breast cancer through media

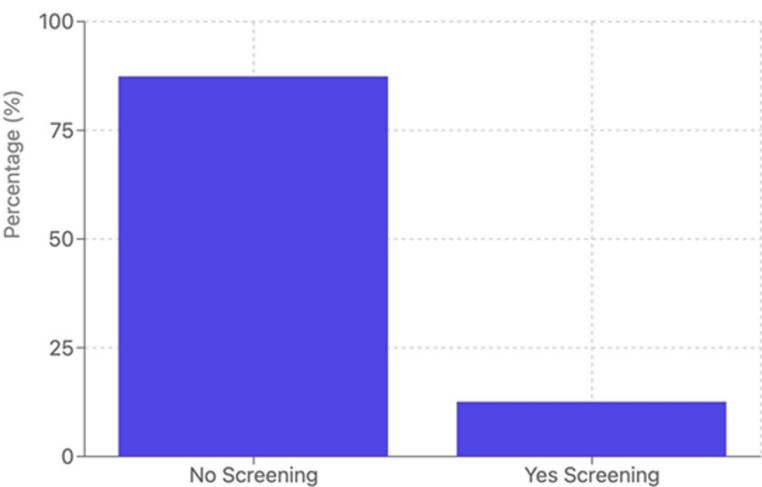

**Fig 1. Distribution of breast cancer screening uptakes (N =985).**

programs. This is similar to research done in Ethiopia where 57.91% heard of breast cancer through Television/Radio [17] and in Addis Abba, 70.10% heard about breast cancer from TV/Radio [18]. The Directorate of Health Promotion and Education under the Ministry of Health of The Gambia is a media-friendly center that is responsive to current health issues and proactive in educating the population on common health problems in the country.

The respondents demonstrated an average awareness level of breast cancer risk factors. They correctly identified obesity/overweight (35.2%), female sex (35.0%), family history (29.0%), and cigarette smoking/alcohol consumption (22.4%) as risk factors for breast cancer. This finding is similar to that of a study conducted by (Ramathuba et al., 2015) in South Africa that showed the use of oral contraceptives and late menopause as less common risk factors. Similarities were also found in a study in Nigeria, where 11.8% and 18.7% of respondents reported late menopause and birth control pill use, respectively, as risk factors for breast cancer [19]. The country has low contraceptive uptake among reproductive women (19%) [13]; as such, respondents have limited knowledge of oral contraceptives as a risk factor for breast cancer. Conversely, knowledge about late menopause as a risk factor is lower than that reported in many studies in sub-Saharan Africa [9,17,20–22].

This study identified a large awareness gap regarding breast cancer screening methods and diagnostics. Less than 15% of the respondents knew about clinical breast examinations, 10% knew about breast self-examinations, and up to 4% knew about mammography. This finding corroborates the spatial availability of breast cancer diagnostic services in the country, as reported by [12]. Efforts should be centered on improving breast cancer education, including knowledge of self-breast examinations and diagnostic methods that could improve women's screening practices. Breast cancer pre-vention among women is a global target desirable for all stakeholders. The respondents advanced the initiation of breast-feeding (38.8%) and improved physical exercise (37.8%) as methods for preventing breast cancer in women. These are among the WHO-recommended preventive methods for breast cancer in women [3].

The respondents' awareness of the tell-tale signs of the symptoms of breast cancer aligned around four major signs: swelling of the breast (48.2%), nipple pain(45.2%), breast lump (37.1%), and sores on the breast (26.8%). These findings are similar to those of studies conducted among women in Qatar, Southwest Ethiopia, Uganda, Cameroon, India, Nigeria and Northwest Ethiopia [17,21–26]. However, there was limited awareness among the respondents regarding the signs of nipple discharge, retracted nipple, change in breast color, scaling, and thickening of the nipple, and efforts are needed

**Table 2. Breast cancer awareness of respondents.**

| variable | n | % |
| --- | --- | --- |
| **Heard about breast cancer (n=985)** | | |
| Yes | 864 | 87.7 |
| No | 121 | 12.3 |
| **Source of information*** | | |
| Family member | 291 | 29.5 |
| Health worker | 305 | 31.0 |
| Teacher | 91 | 9.2 |
| Friends | 277 | 28.1 |
| mass media | 407 | 41.3 |
| Others | 48 | 4.9 |
| **Knew risk factors of breast cancer*** | | |
| Obesity/Overweight | 125 | 35.2 |
| Old age | 164 | 16.6 |
| Family history of breast cancer | 293 | 29.0 |
| Birth of first child after the age of 30 years | 110 | 11.2 |
| Early onset of menses (before age 12) | 83 | 8.4 |
| Late menopause (after 55 years) | 49 | 5.0 |
| Late initiation of breastfeeding | 121 | 12.3 |
| Being a woman | 345 | 35.0 |
| Cigarettee smoking/Alcohol consumption | 221 | 22.4 |
| Use of Oral contraceptive | 82 | 8.3 |
| Exposure to radiation | 235 | 23.2 |
| **Knew breast examination methods*** | | |
| Breast self-examination | 101 | 10.5 |
| Clinical breast examination | 147 | 14.5 |
| Mammography | 36 | 3.6 |
| **Knew the preventive methods of breast cancer risk*** | | |
| Improve physical exercise | 372 | 37.8 |
| Initiation of breast feeding | 382 | 38.8 |
| Limit Alcohol intake | 235 | 23.8 |
| Avoid hormonal replacement therapy | 115 | 11.7 |
| **Knew who should be screen for breast cancer*** | | |
| Older women | 404 | 41.0 |
| Adolescents | 619 | 62.8 |
| Women in reproductive age | 747 | 73.8 |
| Pregnant women | 485 | 49.2 |

*Multiple responses

to raise awareness. These findings highlight the complex interplay of structural, knowledge-based, and personal barriers influencing breast cancer screening uptake in this rural population, suggesting the need for targeted interventions to address these multiple barriers to screening.

The uptake of screening for breast cancer was low among respondents, as less than 13% had ever undergone any form of clinical breast examination. Among those who had been screened for breast cancer, more than 62% done clinical breast examination. Mammography was the least conducted (1.1%) breast cancer screening method among the

**Table 3. Distribution of breast cancer screening uptake.**

| Variables | n | % |
|---|---|---|
| Which breast screening method? (n=124)* | | |
| Breast self-examination | 49 | 39.5 |
| Clinical breast examination | 78 | 62.6 |
| Mammography | 14 | 1.1 |
| Reasons for breast screening* | | |
| Breast pain | 124 | 100 |
| Notice a breast lump | 50 | 40.3 |
| Advice from a health worker | 73 | 58.9 |
| Reasons for not breast screening (n=861)* | | |
| Lack of knowledge | 505 | 58.7 |
| Forgetfulness | 99 | 11.5 |
| Fear of the finding a mass | 97 | 11.3 |
| Service unavailability | 116 | 13.5 |
| High cost | 113 | 13.1 |

*Multiple responses

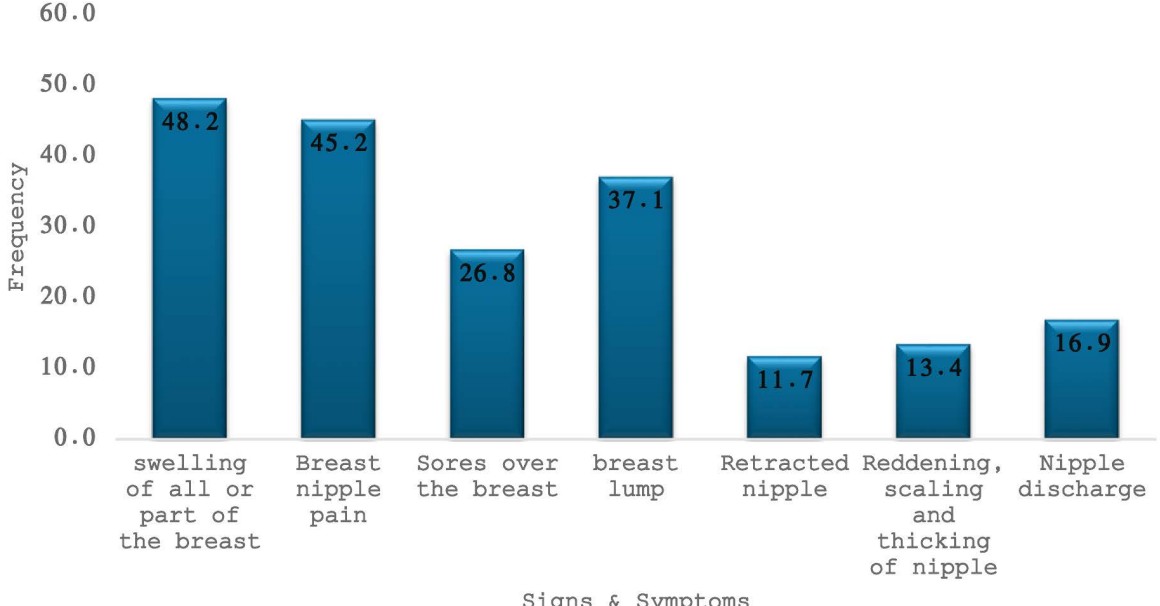

**Fig 2. Distribution of breast cancer's signs and symptoms.**

respondents because of the unavailability of this screening method in rural areas, as evidenced by Sanyang et al., 2021, who found that breast cancer histopathological diagnosis and surgical management remain unavailable to over 50% of the population in The Gambia [12]. Breast Self-Examination, which is a simple and cost-free method of screening for breast diagnosis, was conducted by less than 40% of respondents. These findings are lower than those reported in studies conducted in Uganda (76.5%), Cameroon (47.0%), Northwest Ethiopia (45.8%), and Addis Ababa (43.6%) [18,21–23].

**Table 4. Adjusted Odds Ratios of breast cancer screening uptake by women's selected characteristics.**

| Variables | aOR | 95%CI | p-value |
|---|---|---|---|
| **Ethnicity** | | | |
| Mandinka *(ref)* | 1 | | |
| Fula | 1.552 | 0.872 - 2.761 | 0.135 |
| Wolof | 1.559 | 0.977 - 2.486 | 0.062 |
| Serahuli | 1.359 | 0.541 - 3.387 | 0.517 |
| Jola | 0.867 | 0.188 - 3.990 | 0.854 |
| Serer | 2.821 | 0.521 - 15.267 | 0.229 |
| **Parity** | | | |
| Nullipara *(ref)* | 1 | | |
| Primipara | 0.12 | 0.10 - 1.448 | 0.095 |
| Two | 0.061 | 0.005 - 0.791 | 0.032* |
| Three | 0.075 | 0.006 - 0.967 | 0.047* |
| Four | 0.238 | 0.019 - 3.021 | 0.269 |
| Five | 0.065 | 0.005 - 0.877 | 0.040* |
| Six & Above (Multipara) | 0.196 | 0.015 - 2.513 | 0.211 |
| **Current Occupation** | | | |
| Unemployed *(ref)* | 1 | | |
| Civil servant | 2.778 | 1.174 - 6.573 | 0.020* |
| Student | 3.111 | 1.453 - 6.663 | 0.003* |
| Housewife | 1.022 | 0.489 - 2.135 | 0.954 |
| Farmer | 0.936 | 0.413 - 2.122 | 0.875 |
| Business | 1.138 | 0.506 - 2.560 | 0.755 |

*Statistical significance p<0.05, aOR, Adjusted Odds Ratio

However, the uptake of breast self-examination is higher than that in a similar study conducted in the southern parts of Ethiopia (21.1%) [27]. Variability in breast self-examination among countries could be associated with differences in social literacy. Eventually, breast pain was one of the reasons for uptake, and a lack of awareness was a major barrier to breast cancer screening in this study.

This study's findings reveal a consistent inverse relationship between parity and breast cancer screening uptake, suggesting that women with children face unique barriers to accessing preventive healthcare services. This pattern might be attributed to competing priorities of childcare responsibilities, time constraints, and resource allocation within families. The decreased likelihood of screening among multiparous women highlights the need for targeted interventions that address the specific challenges faced by mothers, such as mobile screening services or community-based programs that integrate childcare support. Occupational status emerged as a crucial determinant of screening practices, with formal employment and educational engagement positively influencing screening behaviors. The higher screening rates among civil servants and students likely reflect a combination of factors including better health literacy, access to information, financial resources, and possibly health insurance coverage through employment. Unlike previous studies where education alone predicted screening uptake [23,27], our findings suggest that employment status and its associated benefits may be more influential in the rural Gambian context. This underscores the importance of workplace health programs and the need to extend similar advantages to women in informal employment sectors.

## Strengths and limitations

### Strengths

This study had several strengths. First, it is one of the few community-based studies to examine breast cancer screening uptake among rural women in The Gambia, providing crucial insights into healthcare-seeking behaviors in underserved populations. The high response rate (90.2%) and the inclusion of hard-to-reach communities enhanced the representativeness of the study. Additionally, the findings may be generalizable to other rural regions of The Gambia given the socio-demographic similarities between the northern and southern regions. The use of validated data collection tools and trained interviewers who could communicate in local languages further strengthens the validity of our findings.

### Limitations

Several limitations of this study should be considered when interpreting our results. First, the cross-sectional design precludes the establishment of causal relationships between breast cancer screening uptake and the associated factors. Second, the self-reported nature of the data may have introduced social desirability and recall bias, potentially affecting the accuracy of screening uptake reports. Third, while we attempted to minimize information bias through validated questionnaires and trained interviewers, language barriers and varying interpretations of screening uptake might have influenced the responses, particularly in communities with low health literacy. Finally, we did not assess the quality and accessibility of the available screening services, which could provide additional information to the observed screening patterns.

## Conclusion

This study provides important insights into breast cancer awareness and screening uptake among rural women in The Gambia. While awareness of breast cancer signs, symptoms, risk factors, and preventive measures was relatively high, screening uptake (12.6%) were substantially lower than the reported rates in other sub-Saharan African settings. The findings revealed significant disparities in screening uptake, with occupation and parity emerging as the key predictors. Specifically, women in formal employment (civil servants) and students showed higher screening rates, while increased parity was associated with a decreased screening likelihood. Despite the high awareness of breast cancer, screening uptake among rural women in The Gambia was notably low, primarily due to limited knowledge of screening methods, financial constraints, and service unavailability.

These findings underscore the urgent need for comprehensive interventions targeting both individual and systemic screening barriers. We recommend that the Ministry of Health and relevant stakeholders strengthen health system capacity by making breast cancer screening services more accessible and affordable in rural areas while implementing targeted awareness programs that address specific barriers faced by different demographic groups. Such strategic interventions could significantly improve screening uptake and ultimately contribute to better breast cancer outcomes in rural women in The Gambia.

## Acknowledgements

We acknowledge the contributions of Solace Foundation volunteers from the Students of Public Health at Gambia College and Medical Students from the University of The Gambia during data collection. Their dedication and services during this period contributed to the success of this research. We also wish to acknowledge the respondents for their active participation during data collection.

## Author contributions

**Conceptualization:** Bakary Kinteh, Fatoumatta Jitteh, Mansour Badjie, Amadou Barrow, Lamin Jaiteh.
**Data curation:** Bakary Kinteh, Mansour Badjie.

**Formal analysis:** Bakary Kinteh, Fatoumatta Jitteh, Mansour Badjie, Amadou Barrow.

**Investigation:** Bakary Kinteh, Lamin Jaiteh.

**Methodology:** Bakary Kinteh, Fatoumatta Jitteh, Mansour Badjie, Amadou Barrow, Lamin Jaiteh.

**Software:** Amadou Barrow.

**Supervision:** Bakary Kinteh, Mansour Badjie.

**Validation:** Bakary Kinteh.

**Visualization:** Bakary Kinteh.

**Writing – original draft:** Bakary Kinteh, Fatoumatta Jitteh, Mansour Badjie, Amadou Barrow, Lamin Jaiteh.

**Writing – review & editing:** Bakary Kinteh, Amadou Barrow, Lamin Jaiteh.

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
