## [Decision Letter · Decision Letter 0]

5 Nov 2024

PONE-D-24-19264Exploring Breast Cancer Awareness and Screening Practices Amongst Gambian Women in the Gambia; Community-based- cross-sectional studyPLOS ONE

Dear Dr. Kinteh,

Thank you for submitting your manuscript to PLOS ONE. After careful consideration, we feel that it has merit but does not fully meet PLOS ONE’s publication criteria as it currently stands. Therefore, we invite you to submit a revised version of the manuscript that addresses the points raised during the review process.

We look forward to receiving your revised manuscript.

Kind regards,

Denekew Bitew Belay, Ph.D

Academic Editor

PLOS ONE

2. In the online submission form, you indicated that [The data used to support the findings of this study are available from the Corresponding author upon reasonable request at: bakarykinteh461@gmail.com]. All PLOS journals now require all data underlying the findings described in their manuscript to be freely available to other researchers, either 1. In a public repository, 2. Within the manuscript itself, or 3. Uploaded as supplementary information. This policy applies to all data except where public deposition would breach compliance with the protocol approved by your research ethics board. If your data cannot be made publicly available for ethical or legal reasons (e.g., public availability would compromise patient privacy), please explain your reasons on resubmission and your exemption request will be escalated for approval.

Additional Editor Comments (if provided):

The authors used the two different terms in the study "multivariate logistic regression” and "multivariable logistic regression “interchangeably. Please use the later term consistently.

The authors used "multivariable logistic regression” and use this term throughout. The overall writings should be revised.

Reviewers' comments:

Reviewer's Responses to Questions

**Comments to the Author**

1. Is the manuscript technically sound, and do the data support the conclusions?

Reviewer #1: Partly

Reviewer #2: Yes

Reviewer #3: Yes

Reviewer #4: Partly

2. Has the statistical analysis been performed appropriately and rigorously? 

Reviewer #1: Yes

Reviewer #2: Yes

Reviewer #3: Yes

Reviewer #4: No

3. Have the authors made all data underlying the findings in their manuscript fully available?

Reviewer #1: Yes

Reviewer #2: Yes

Reviewer #3: Yes

Reviewer #4: Yes

4. Is the manuscript presented in an intelligible fashion and written in standard English?

Reviewer #1: Yes

Reviewer #2: Yes

Reviewer #3: No

Reviewer #4: No

5. Review Comments to the Author

Reviewer #1: The authors have described findings from of breast cancer awareness and screening practices among rural Gambian women in a community-based cross-sectional study. Although awareness was high with 87.7% of women being aware of breast cancer, only 13% practised screening.

Major issues;

1-The distance or proximity of the closest medical facility to the subjects. What kind/level of service are they able to receive, when available?

2-Did cultural or religious practice affect the practice of screening for breast cancer seeing the rather low number (13%) in relation to an 87.7% awareness level of cancer. These were not explored in the study. It will be helpful to give more information on factors that could influence your results such as cultural beliefs, healthcare infrastructure, and accessibility.

3-study design and setting – what health facility is located in this study areas, how accessible are they to study subjects and what infrastructure is on ground to help with breast cancer detection? Providing detailed information on the healthcare infrastructure or specific barriers faced by women in accessing these services will be helpful in appreciating the magnitude of the problem to readers.

Minor issues;

1. What informed the choice of rural women for this study seeing similar study had been done in female undergraduate students in the past?

2. Strengths and limitations (page 11) – although the population is nearly homogeneous for example proximity to health care facility with screening services could affect their screening practices regarding breast cancer.

Reviewer #2: Topic: Exploring Breast Cancer Awareness and Screening Practices amongst Gambian Women in the Gambia; Community-based- cross-sectional study

Dear Editor PLos One Journal,

Many thanks for the opportunity to review this paper from the Gambia. I am grateful.

General Comments

A good and interesting paper. It adds to the breast cancer literature from the Gambia. However, the topic needs revision.

The background outlined the key issues related to breast cancer awareness and screening practices in the Gambia

The statistical analysis were appropriate

The findings presented were relevant and the discussion gave an in-dept explanation on the topic.

Conclusion: the conclusion is generally sound and based on the findings; however, it needs a slight revision.

Specific Comments

1. Tropic: One of the study variables looked at nationality: Gambian and non-Gambino. Therefore, the authors should consider removing the term Gambian on the topic. The proposed topic should read, "EExploring Breast Cancer Awareness and Screening Practices amongst Women in the Gambia: A Community-Based cross-sectional Study.”

2. What was the justification to choose simple random sampling to select participants in households with more than a single eligible respondent on line 130-131? Secondly, how were/were households with no eligible respondent dealt with in the study?

3. Were the research assistant from Solace foundation trained for the data collection at the community level?

4. In Table 1 and table 2, authors were using participants and in other tables respondents. They should be consistent with the terms. Table 2 should read “Breast cancer awareness of respondents.”

5. In lines 202 and 207, the term “our respondents” should be changed to “the respondents” all throughout the document.

6. Table 2: all variables with multiple respondents should be indicated, eg “ Knew the preventive methods of breast cancer risk.”

7.Line 239 Kinteh et al , no year was added to the reference. Similar to Sanyang et al., line 290. Therefore, it should be rephrased

8. Conclusion Line 330-332: “The predictors of breast cancer screening practices were influenced by the parity of the women and the current occupation of the respondents” . Do they mean ethnicity? Parity was not, even though some of its sub categories indicated so.

Reviewer #3: The article provides interesting findings on breast cancer in The Gambia. I think the paper is worthy of publication, but I have a few questions and suggestions for improvement with regard to the methodology and the evaluation strategy.

The authors should better justify the statistical model they chose with mathematical expressions for the models. The descriptive results are not sufficient, charts can be used to support the results.

Although, the article demonstrated scholarly argument, the article is poorly written with a lot of typos and grammatical errors. The paper needs some editing and proof-reading as well as some double-check for consistency in spelling ( I have indicated some out of so many).

Accept with major revision

Reviewer #4: General comment

The study aimed to explore breast cancer awareness and screening practices of rural Gambian women. This is titled “Exploring Breast Cancer Awareness and Screening Practices 1 Amongst Women in The Gambia; Community-based- cross-sectional study”. The study is premised on the limited strategies for addressing breast cancer in The Gambia.

Major comment

Research on breast is laudable. However, the more specific and focused research is, the better. The study focused is inadequately articulated and contextualised. Absence of the operational definition of the main outcome variable of interest is a concern. Nonetheless, I suggest the authors focus solely on breast cancer screening practices.

Specific comments

Title:

• I suggest modification of the title to read as “Barriers and associated characteristics of breast Cancer Screening Practices amongst women in northern part of The Gambia;”

Abstract:

• Background

o Lines 20-21: could be replaced with justification relating to breast cancer screening practices

o Line 22: study objective may thereafter reflect the modified title.

• Methods: consider recasting the statement to reflect the following

o Line 25: Why among rural women?? This is not reflected in the body of the work

o Lines 26-27: authors should recast … interviews may be conducted using structured questionnaires deploy using Kobo toolbox

o Line 28: delete the phrase “SPSS version 28” - Avoid inclusion of software in abstract.

o Line 29: … to determine correlation?? Pearson Chi-square is used to examine association between two categorical variables

o Line 30: replace test with model

o Provide a statement to indicate that adjusted odds ratios was reported in the Results section

• Results:

o Line 34: replace the pronoun “our” with “the”, avoid the use of such pronoun as much as possible

o Lines 36-37: I could not find the result in the body of the work to justified the inclusion of the statement as “risk factors”. However, I guess the authors meant to say respondents’ characteristics

o Lines 39-42: please improve on the reporting style for the statement as only occupation and parity are reported.

• Conclusions:

o Line 43: average of what??? The statement is unclear!

o Line 44: what makes it optimal ... this is not indicated in the result nor in the method section

o Line 45: ... could not be traced to any of the result presented in the Results section of the abstract. Nonetheless, if the suggested title is sustained, it could be appropriate provided that it is linked to the results section.

Introduction:

o Lins 58-59: the statement requires appropriate referencing

o Lines 83-86: the two statements are contradictory; first, lack of knowledge and second, good knowledge

o Line 99 as commented earlier – operational definition of the main outcome of interest is required

o The study justification is rather weak with respect to with the BC awareness or its screening practice among the target population, Most importantly, non-inclusion of the targeted population known practices towards BC screening

Methods:

• Study design and setting

o Lines 107-111: Authors should mention no of districts per the Local government authorities in the northern part of the Gambia. This should be followed by the average numbers of communities per district. Expectedly, the setting should also include the spread of the screening centres

• Sample size determination

o Line 120: the statement “confidence interval of 99% with 50% precision” is unclear! Cross sectional one population study estimated sample size usually utilised within 1 – 5% level of precision. Authors are advised to state the correct parameters used to arrive at 985 as the estimated sample size.

• Sampling strategy

o Lines 123-125: This is not a selection stage.

o Lines 125-131: Questions that ought to have been answered by the “setting” section

Districts selected - out of how any districts?? Total districts in each LGAs should have been mentioned under the study setting

what is the average community per district??

How many households selected per community???

how many participants selected per household ... where more than one was eligible??

• Data collection and management

o Line 132: replace the sub-section title with “Data collection instruments”

o Line 150: it may be necessary to provide justification for using 3 local languages

• Authors are advised to insert a subsection titled “Study variables”. This should clearly state the operational definition of the outcome variable and the selected characteristics of the women.

•

• Statistical analysis

o Line 158: replace the sub-section title with “Data management and analysis”

o Lines 164-165: replace with “All analysis was conducted at a 5% level of significance”

Results:

• insert “Women’s characteristics” as a subsection

• Line 175: How figure (1012) was obtained needs to be clearly stated in the Methods - sampling strategy subsection

• Association of the selected women’s characteristics and breast cancer screening

o Line Technically, the statement “In total, 529 households were included in the study, with 273 children aged 12-59 months enrolled” could be not right. More than a child could come from a household but NOT the other way round.

o Again, there is no table/figure that is referenced

• Table 1

o While combining the distribution of women by their characteristics and the characterises association with breast cancer is welcome, I will suggest that

o 1. Total column should come before the columns for Yes and No for breast cancer screening

o 2. Yes and No columns of breast cancer screening percentages should be presented in row-wise”.

o The use of chi-square becomes inappropriate in situations where there is zero value in any of the contingency table cells

o Authors should ensure all the variables listed in Table 1 are clearly defined under the methods, study variables subsection

o Table 2 – should appear before the presentation of the “Association between the selected women characteristics and the breast cancer screening practice

• Discussion

• The unclear focus of the study, lines 234-237 in particular, is a concern here. Nonetheless, authors are advised to avoid writing the results under the discussion section – see lines 302 – 317

6. PLOS authors have the option to publish the peer review history of their article (what does this mean? ). If published, this will include your full peer review and any attached files.

**Do you want your identity to be public for this peer review?** For information about this choice, including consent withdrawal, please see our Privacy Policy .

Reviewer #1: **Yes: ** Francis Akor

Reviewer #2: No

Reviewer #3: **Yes: ** OLUWAYEMISI OYERONKE ALABA

Reviewer #4: No

---

## [Author Response · Author response to Decision Letter 0]

2 Dec 2024

November 14, 2024

Manuscript ID: [PONE-D-24-19264] - [EMID:b560eba95e797023]

Title: “Exploring Breast Cancer Awareness and Screening Practices Amongst Rural Women in The Gambia: Community-based Cross-sectional Study”

RE: Revised manuscript submission and response to reviewers’ comments

Dear Editor,

This letter is in reference to your email dated November 5, 2024 with reviewers’ comments. We are very pleased that the manuscript is potentially acceptable for publication in PLOS ONE once we have carried out the revisions.

We would like to thank the reviewers for these insightful and helpful comments and for giving us the chance to revise our manuscript. We believe the revised manuscript has been significantly improved and the reviewers’ comments have been addressed adequately. We think in its current form it will make a valuable contribution to the literature on this increasingly important topic.

Please find for your kind consideration the following:

• A section-by-section response to the comments and suggestions of the reviewers (below).

• The revised manuscript, provided as a marked-up copy and a clean copy.

We hope that these changes meet with your favourable consideration. Please do not hesitate to get in touch if you require any further information.

Bakary Kinteh

Corresponding Author

Reviewer #1:

The authors have described findings from of breast cancer awareness and screening practices among rural Gambian women in a community-based cross-sectional study. Although awareness was high with 87.7% of women being aware of breast cancer, only 13% practised screening.

Major issues;

1-The distance or proximity of the closest medical facility to the subjects. What kind/level of service are they able to receive, when available?

Response: The specifics of available services are beyond the scope of this paper. We appreciate your insights on this and this is clearly acknowledged as a limitation of this study. We assume that assessing the quality and accessibility of available screening services could be helpful in understanding the observed screening patterns.

2-Did cultural or religious practice affect the practice of screening for breast cancer seeing the rather low number (13%) in relation to an 87.7% awareness level of cancer. These were not explored in the study. It will be helpful to give more information on factors that could influence your results such as cultural beliefs, healthcare infrastructure, and accessibility.

Response: You are right we could account for additional potential confounders and even effect modifiers by including variables such as religion, cultural beliefs, healthcare infrastructure and social support systems. These could be future research directions for research in this domain of women’s health. Thank you for bringing this to our attention.

3-study design and setting – what health facility is located in this study areas, how accessible are they to study subjects and what infrastructure is on ground to help with breast cancer detection? Providing detailed information on the healthcare infrastructure or specific barriers faced by women in accessing these services will be helpful in appreciating the magnitude of the problem to readers.

Response: We have now revised the study design and setting section as suggested. Thank you.

Minor issues;

1. What informed the choice of rural women for this study seeing similar study had been done in female undergraduate students in the past?

Response: Our decision was informed by several critical factors. While previous studies among female university students provided valuable insights into breast cancer awareness in an educated, urban population, rural women (constituting approximately 55- 60% of The Gambia's female population) face unique challenges including geographical isolation, limited healthcare infrastructure, and financial constraints. Moreover, rural women often present with late-stage breast cancer due to delayed healthcare seeking, making them a crucial yet understudied population. Understanding their screening practices and barriers is essential for developing targeted interventions and informing health policies aimed at reducing rural-urban health disparities. Therefore, this study complements, rather than duplicates, previous research by addressing a critical knowledge gap in breast cancer screening practices among a vulnerable and underserved population.

2. Strengths and limitations (page 11) – although the population is nearly homogeneous for example proximity to health care facility with screening services could affect their screening practices regarding breast cancer.

Response: Thank you for the insights and we have now expanded on our study limitation as expected.

Reviewer #2:

Topic: Exploring Breast Cancer Awareness and Screening Practices amongst Gambian Women in the Gambia; Community-based- cross-sectional study

Dear Editor PLos One Journal,

Many thanks for the opportunity to review this paper from the Gambia. I am grateful.

General Comments

A good and interesting paper. It adds to the breast cancer literature from the Gambia. However, the topic needs revision.

The background outlined the key issues related to breast cancer awareness and screening practices in the Gambia

The statistical analysis were appropriate

The findings presented were relevant and the discussion gave an in-dept explanation on the topic.

Conclusion: the conclusion is generally sound and based on the findings; however, it needs a slight revision.

Response: We appreciate your insightful comment, Thank you.

Specific Comments

1. Topic: One of the study variables looked at nationality: Gambian and non-Gambian. Therefore, the authors should consider removing the term Gambian on the topic. The proposed topic should read, "Exploring Breast Cancer Awareness and Screening Practices amongst Women in the Gambia: A Community-Based cross-sectional Study.”

Response: We now updated the title as suggested and we have also improved our study’s eligibility criteria to adequately address our inclusion and exclusion criteria.

2. What was the justification to choose simple random sampling to select participants in households with more than a single eligible respondent on line 130-131? Secondly, how were/were households with no eligible respondent dealt with in the study?

Response: Simple random sampling was chosen for selecting participants within households with multiple eligible respondents to ensure equal probability of selection and minimize selection bias that could arise from convenience sampling or voluntary participation. This method helped prevent overrepresentation of more available household members (e.g., unemployed or housewives) who might have different screening behaviors. Regarding households without eligible respondents, these were documented and factored into our response rate calculations to maintain methodological transparency, with replacements systematically selected from the next household in a predetermined direction until we achieved our target sample size. We have reflected these strategies in the manuscript as suggested. Thank you.

3. Were the research assistant from Solace foundation trained for the data collection at the community level?

Response: Yes, all research assistants underwent standardized training conducted by the Solace Foundation, which included instruction on data collection using the Kobo Toolbox, interviewing techniques, and translation protocols for three major local languages (Mandinka, Wolof, and Fula). Prior to field deployment, the trained research assistants participated in a pilot study in the West Coast Region to ensure proficiency in data collection procedures and consistency in questionnaire administration.

4. In Table 1 and table 2, authors were using participants and in other tables respondents. They should be consistent with the terms. Table 2 should read “Breast cancer awareness of respondents.”

Response: We now change all of them to respondents as suggested.

5. In lines 202 and 207, the term “our respondents” should be changed to “the respondents” all throughout the document.

Response: We now change all of them to respondents as suggested.

6. Table 2: all variables with multiple respondents should be indicated, eg “ Knew the preventive methods of breast cancer risk.”

Response: This is implemented as suggested. Thank you.

7.Line 239 Kinteh et al , no year was added to the reference. Similar to Sanyang et al., line 290. Therefore, it should be rephrased

Response: We provide the year and rephrased accordingly. Thank you.

8. Conclusion Line 330-332: “The predictors of breast cancer screening practices were influenced by the parity of the women and the current occupation of the respondents” . Do they mean ethnicity? Parity was not, even though some of its sub categories indicated so.

Response: Thank you for this important clarification. Our conclusion correctly identifies occupation and parity categories (specifically women with two, three, and five children) as significant predictors of breast cancer screening practices, not ethnicity. While ethnicity showed some variations in screening uptake, none were statistically significant (p>0.05). Thank you.

Reviewer #3:

The article provides interesting findings on breast cancer in The Gambia. I think the paper is worthy of publication, but I have a few questions and suggestions for improvement with regard to the methodology and the evaluation strategy.

Response: Thank you for those comments.

The authors should better justify the statistical model they chose with mathematical expressions for the models. The descriptive results are not sufficient, charts can be used to support the results.

Response: We appreciate your suggestion for enhanced statistical presentation. We have now included a detailed mathematical expression of our binary logistic regression model, where logit(p) = ln(p/1-p) = β₀ + β₁X₁ + β₂X₂ + ... + βₖXₖ, with p representing the probability of breast cancer screening practice and β coefficients representing the effects of our predictor variables. This model was chosen due to the binary nature of our outcome variable (screening: yes/no) and its ability to estimate adjusted odds ratios while controlling for multiple predictors simultaneously. To enhance visual interpretation of our results, we have also added a bar chart depicting breast cancer screening practices distribution and another visualizing the main barriers to screening, which complement our tabulated findings and provide a clearer representation of the patterns observed in our data.

Although, the article demonstrated scholarly argument, the article is poorly written with a lot of typos and grammatical errors. The paper needs some editing and proof-reading as well as some double-check for consistency in spelling ( I have indicated some out of so many).

Accept with major revision

Response: We now revised the manuscript and addressed typos and grammatical errors accordingly. Thank you.

Reviewer #4:

General comment

The study aimed to explore breast cancer awareness and screening practices of rural Gambian women. This is titled “Exploring Breast Cancer Awareness and Screening Practices 1 Amongst Women in The Gambia; Community-based- cross-sectional study”. The study is premised on the limited strategies for addressing breast cancer in The Gambia.

Response: Thank you for the comment.

Major comment

Research on breast is laudable. However, the more specific and focused research is, the better. The study focused is inadequately articulated and contextualised. Absence of the operational definition of the main outcome variable of interest is a concern. Nonetheless, I suggest the authors focus solely on breast cancer screening practices.

Response: We have carefully revised the manuscript to sharpen its focus and context, specifically examining both breast cancer awareness and screening practices among rural women in The Gambia. The dual focus is intentional and necessary, as understanding awareness levels provides crucial context for interpreting screening behaviors, particularly in rural settings where health literacy and healthcare access intersect. We have now included clear operational definitions of our outcome variables, with breast cancer screening practice defined as "ever having undergone any form of breast cancer screening (breast self-examination, clinical breast examination, or mammography)." Throughout the manuscript, we have maintained consistent attention to both awareness and screening practices, as these are inextricably linked in the Gambian rural context, where understanding the relationship between knowledge and preventive health behaviors is crucial for developing effective interventions. This approach allows for a comprehensive understanding of the barriers to screening uptake while maintaining a focused analysis of screening practices as our primary outcome of interest.

Specific comments

Title:

• I suggest modification of the title to read as “Barriers and associated characteristics of breast Cancer Screening Practices amongst women in northern part of The Gambia;”

Response: We appreciate your suggested title modification. However, we respectfully prefer to maintain our current title "Exploring Breast Cancer Awareness and Screening Practices Amongst Rural Women in The Gambia: Community-based Cross-sectional Study" as it more accurately reflects our study's comprehensive scope, methodology, and findings as suggested by reviewer 2. While barriers and associated characteristics are important components of our analysis, our study examined both awareness and screening practices as interconnected aspects of breast cancer prevention in rural communities. Additionally, our current title explicitly identifies the study design and target population, providing immediate context for readers, and aligns with standard epidemiological reporting guidelines for observational studies.

Abstract:

• Background

o Lines 20-21: could be replaced with justification relating to breast cancer screening practices

o Line 22: study objective may thereafter reflect the modified title.

Response: We have now revised our abstract as suggested. Thank you

• Methods: consider recasting the statement to reflect the following

o Line 25: Why among rural women?? This is not reflected in the body of the work

o Lines 26-27: authors should recast … interviews may be conducted using structured questionnaires deploy using Kobo toolboxo Line 28: delete the phrase “SPSS version 28” - Avoid inclusion of software in abstract.

Response: Implemented as suggested in our revised version. Thank you

o Line 29: … to determine correlation?? Pearson Chi-square is used to examine association between two categorical variables

o Line 30: replace test with model

o Provide a statement to indicate that adjusted odds ratios was reported in the Results section

Response: We have revised and addressed our abstract’s method section accordingly. Thank you.

• Results:

o Line 34: replace the pronoun “our” with “the”, avoid the use of such pronoun as much as possible

o Lines 36-37: I could not find the result in the body of the work to justified the inclusion of the statement as “risk factors”. However, I guess the authors meant to say respondents’ characteristics

o Lines 39-42: please improve on the reporting style for the statement as only occupation and parity are reported.

Response: Implemented as suggested in our revised version. Thank you

• Conclusions:

o Line 43: average of what??? The statement is unclear!

o Line 44: what makes it optimal ... this is not indicated in the result nor in the method section

o Line 45: ... could not be traced to any of the result presented in the Results section of the abstract. Nonetheless, if the suggested title is sustained, it could be appropriate provided that it is linked to the results section.

Response: We have revised our conclusion and all your concerns are adequately addressed. Thank you

Introduction:

o Lins 58-59: the statement requires appropriate referencing

o Lines 83-86: the two statements are contradictory; first, lack of knowledge and second, good knowledge

o Line 99

---

## [Decision Letter · Decision Letter 1]

2 Jan 2025

PONE-D-24-19264R1Exploring Breast Cancer Awareness and Screening Practices Amongst Rural Women in The Gambia: Community-based Cross-sectional StudyPLOS ONE

Dear Dr. Kinteh,

Thank you for submitting your manuscript to PLOS ONE. After careful consideration, we feel that it has merit but does not fully meet PLOS ONE’s publication criteria as it currently stands. Therefore, we invite you to submit a revised version of the manuscript that addresses the points raised during the review process.

We look forward to receiving your revised manuscript.

Kind regards,

Denekew Bitew Belay, Ph.D

Academic Editor

PLOS ONE

Journal Requirements:

Reviewers' comments:

Reviewer's Responses to Questions

**Comments to the Author**

1. If the authors have adequately addressed your comments raised in a previous round of review and you feel that this manuscript is now acceptable for publication, you may indicate that here to bypass the “Comments to the Author” section, enter your conflict of interest statement in the “Confidential to Editor” section, and submit your "Accept" recommendation.

Reviewer #1: All comments have been addressed

Reviewer #2: All comments have been addressed

Reviewer #4: (No Response)

2. Is the manuscript technically sound, and do the data support the conclusions?

Reviewer #1: Yes

Reviewer #2: Yes

Reviewer #4: (No Response)

3. Has the statistical analysis been performed appropriately and rigorously? 

Reviewer #1: Yes

Reviewer #2: Yes

Reviewer #4: (No Response)

4. Have the authors made all data underlying the findings in their manuscript fully available?

Reviewer #1: Yes

Reviewer #2: Yes

Reviewer #4: (No Response)

5. Is the manuscript presented in an intelligible fashion and written in standard English?

Reviewer #1: Yes

Reviewer #2: Yes

Reviewer #4: (No Response)

6. Review Comments to the Author

Reviewer #1: Well done for the responses to the queries and with the modifications as suggested in the reviews. No further comments from me

Reviewer #2: No further comments. All comments and concerns have been adequately addressed by the authors.

The paper adds valuable literature of breast cancer awareness to the Gambia.

Reviewer #4: (No Response)

7. PLOS authors have the option to publish the peer review history of their article (what does this mean? ). If published, this will include your full peer review and any attached files.

**Do you want your identity to be public for this peer review?** For information about this choice, including consent withdrawal, please see our Privacy Policy .

Reviewer #1: **Yes: ** Francis Akor

Reviewer #2: **Yes: ** Jainaba Sey-Sawo

Reviewer #4: No

---

## [Author Response · Author response to Decision Letter 1]

10 Jan 2025

Our reference list is maintained. Papers cited are not retracted.

---

## [Decision Letter · Decision Letter 2]

14 Jan 2025

PONE-D-24-19264R2Exploring Breast Cancer Awareness and Screening Practices Amongst Rural Women in The Gambia: Community-based Cross-sectional StudyPLOS ONE

Dear Dr. Kinteh,

Thank you for submitting your manuscript to PLOS ONE. After careful consideration, we feel that it has merit but does not fully meet PLOS ONE’s publication criteria as it currently stands. Therefore, we invite you to submit a revised version of the manuscript that addresses the points raised during the review process.

We look forward to receiving your revised manuscript.

Kind regards,

Denekew Bitew Belay, Ph.D

Academic Editor

PLOS ONE

Journal Requirements:

Additional Editor Comments:

The reviewer’s comments and concerns need to be carefully addressed.

Reviewers' comments:

Reviewer's Responses to Questions

**Comments to the Author**

1. If the authors have adequately addressed your comments raised in a previous round of review and you feel that this manuscript is now acceptable for publication, you may indicate that here to bypass the “Comments to the Author” section, enter your conflict of interest statement in the “Confidential to Editor” section, and submit your "Accept" recommendation.

Reviewer #4: (No Response)

2. Is the manuscript technically sound, and do the data support the conclusions?

Reviewer #4: (No Response)

3. Has the statistical analysis been performed appropriately and rigorously? 

Reviewer #4: (No Response)

4. Have the authors made all data underlying the findings in their manuscript fully available?

Reviewer #4: (No Response)

5. Is the manuscript presented in an intelligible fashion and written in standard English?

Reviewer #4: (No Response)

6. Review Comments to the Author

Reviewer #4: The comment is attached!

The responses to the comments are appreciated. However, I would like to see the following comments addressed in order to further enriched the manuscript.

Specific comments

Abstract:

• Background:

o Lines 21-23: replaced with the statement “This study aimed to assess breast

cancer awareness and identify associated factors influencing screening uptake among rural women in The Gambia”

• Methods:

o Line 25: replace the word “Gambia” with “The Gambia”, and elsewhere in the manuscript

o Line 29: replace the phrase “Chi-squared and/or Fisher’s exact test” with “Chi-squared or Fisher’s exact test, as appropriate”

o Line 29: replace the phrase “were reported was in the results.” with “were reported”

• Results:

o Line 36: replace the phrase “Multivariable analysis revealed that students” with “Students”

• Conclusions:

o Lines 43-44: Authors are advised to include result(s) suggesting the phrase “primarily due to limited knowledge of the urgent need for accessible and affordable screening services” in the Results section of the abstract

o Lines 44-45: Let the recommendation statement be presented in a separate sentence.

Material and Methods:

• Study design and setting

o Lines 94: the phrase … northern region of … should be addressed as commented earlier

o Line 105: I suggest the deletion of the phrase “Healthcare Access and Barriers”

o Lines 105-115: There is a need to reference some of the factual statements in the paragraph

• Study population and eligibility criteria

o Line 119: replace with “Study population”

• Sample size determination

o Lines 146-148: Authors should ensure the correctness of the following stated figures!

1. Estimated minimum sample size becomes 1103 (not 1092 as indicated) if adjusted for 10% non-response with 1.5 design effect

2. Besides, let’s assume 1092 is correct! Then, only of 1012 (not 1092) that 97.3% response rate would yield 985 completed the study

• Sampling strategy

o Lines 151-162: replace with “A four-stage sampling method was employed. First, the northern region was purposively selected due to its geographical location, high population density of underserved rural women, and absence of mammography services. This selection aligns with the study's aim of understanding screening uptake in rural, as it represents a significant proportion of rural communities with limited healthcare access. The northern region of The Gambia comprises of Kerewan and Kuntaur local government areas (LGAs), with four (Upper Baddibu, Lower Baddibu, Central Baddibu, and Jokadu) and three (Upper Saloum, Lower Saloum, and Niani) districts, respectively. Second, five communities were randomly selected from each district, yielding 35 study communities in total. Third, within each selected community, households were sampled using probability proportional to size, selecting an average of 25-35 households per community. Finally, one eligible woman was selected from each household.”

o Also important to note, selection of 25-30 households in each of the 35 communities would not yield the minimum sample size estimated (30*35=1050; this cannot lead to 1092 assuming its correct). However, 25-35 households per community will do!

• Data management and statistical analysis

o Lines 197-199: The women age was not covered with the present statement. I refer the authors to my previous advice to replace the statement with “Descriptive statistics such as means with standard deviations (SD) for continuous variables, and frequencies and percentages for categorical variables were used present the women characteristics including their breast cancer awareness and screening practices”.

o Line 199: replace “We employed Pearson's chi-squared and/or Fisher’s exact test to” with “As appropriate, Pearson's chi-squared or Fisher’s exact test was used to … ”

o Lines 204-207: Authors are advised to place the statement “Despite its non-significance at the bivariate analysis level, ethnicity was retained in the model owing to its theoretical importance and our specific interest in understanding its effects on breast cancer screening uptake in relation to other predictors” immediate on line 211 after “bivariate level.” to facilitate a smooth narrative.

Results:

• Figure 1:

o Line 251: replace “practices” with “uptake”

• Table 1

o For sake of clarity to the potential Plos One readers, my previous suggestion is necessary!

Column 2 to be presented as Total (column percentage), while others remain as they were

By so doing, statements on women’s characteristic in lines 240-247 could be referenced adequately.

o Additionally, I suggest that the title should read as “Distribution of women’s socio-demographic characteristics by breast cancer screening uptake

• Lines 291: I suggest the paragraph subsection title be read as “Breast cancer screening practices” as this includes BC signs and symptoms.

• Lines 299-302: The statement “These findings highlight the complex interplay of structural, knowledge-based, and personal barriers influencing breast cancer screening uptake in this rural population, suggesting the need for targeted interventions to address these multiple barriers to screening.” should be moved under Discussion section

Thank you!

7. PLOS authors have the option to publish the peer review history of their article (what does this mean? ). If published, this will include your full peer review and any attached files.

**Do you want your identity to be public for this peer review?** For information about this choice, including consent withdrawal, please see our Privacy Policy .

Reviewer #4: **Yes: ** Rotimi Felix Afolabi

---

## [Author Response · Author response to Decision Letter 2]

23 Jan 2025

We have realized that an article published by Ramya Ahmad et al, 2019 is a retracted paper and removed from our citation and references list.

---

## [Decision Letter · Decision Letter 3]

16 Feb 2025

PONE-D-24-19264R3Exploring Breast Cancer Awareness and Screening Practices Amongst Rural Women in The Gambia: Community-based Cross-sectional StudyPLOS ONE

Dear Dr.  Kinteh,

Thank you for submitting your manuscript to PLOS ONE. After careful consideration, we feel that it has merit but does not fully meet PLOS ONE’s publication criteria as it currently stands. Therefore, we invite you to submit a revised version of the manuscript that addresses the points raised during the review process.

We look forward to receiving your revised manuscript.

Kind regards,

Denekew Bitew Belay, Ph.D

Academic Editor

PLOS ONE

Journal Requirements:

Reviewers' comments:

Reviewer's Responses to Questions

**Comments to the Author**

1. If the authors have adequately addressed your comments raised in a previous round of review and you feel that this manuscript is now acceptable for publication, you may indicate that here to bypass the “Comments to the Author” section, enter your conflict of interest statement in the “Confidential to Editor” section, and submit your "Accept" recommendation.

Reviewer #4: (No Response)

2. Is the manuscript technically sound, and do the data support the conclusions?

Reviewer #4: (No Response)

3. Has the statistical analysis been performed appropriately and rigorously? 

Reviewer #4: (No Response)

4. Have the authors made all data underlying the findings in their manuscript fully available?

Reviewer #4: (No Response)

5. Is the manuscript presented in an intelligible fashion and written in standard English?

Reviewer #4: (No Response)

6. Review Comments to the Author

Reviewer #4: The authors are commended for their efforts in improving this manuscript. The revisions are largely satisfactory; however, a few comments appear to have been addressed inappropriately, likely due to a misunderstanding of their intent. To ensure clarity, I have relisted those comments below for your reference

Abstract:

• Conclusions:

o Lines 43-44: Authors are advised to include result(s) suggesting the phrase “primarily due to limited knowledge of the urgent need for accessible and affordable screening services” in the Results section of the abstract

• The comment requests that the authors report the reasons for the poor uptake of breast cancer screening. Simply referencing “limited knowledge” without presenting supporting data in the Results section would be inappropriate.

• Besides, it is not necessary to include those percentages under the Conclusion section of the abstract. It is better reported under the Results section.

Sample size determination

• Lines 146-148: Authors should ensure the correctness of the following stated figures!

1. Estimated minimum sample size becomes 1103 (not 1092 as indicated) if adjusted for 10% non-response with 1.5 design effect

2. Besides, let’s assume 1092 is correct! Then, only of 1012 (not 1092) that 97.3% response rate would yield 985 completed the study

• Regardless of the statistical software used for sample size computation, the minimum sample size of 662 is accurate based on the presented indices. However, when accounting for the design effect, the minimum sample size increases to 993 (662 * 1.5). If we then adjust for a 10% non-response rate, the required sample size becomes 1103 (993 * 0.9).

• Authors are requested to address this appropriately!

Results:

• Table 1

o For sake of clarity to the potential Plos One readers, my previous suggestion is necessary!

Column 2 to be presented as Total (column percentage), while others remain as they were

By so doing, statements on women’s characteristic in lines 240-247 could be referenced adequately.

• The comments request that the authors include column percentages for the Total column. To avoid any further confusion, I suggest renaming the Total column heading to “n (%)” so that the reported percentages are clearly indicated on a column-wise basis (NOT 100, as presently reported).

Thank you!

7. PLOS authors have the option to publish the peer review history of their article (what does this mean? ). If published, this will include your full peer review and any attached files.

**Do you want your identity to be public for this peer review?** For information about this choice, including consent withdrawal, please see our Privacy Policy .

Reviewer #4: **Yes: ** Rotimi Felix Afolabi

---

## [Author Response · Author response to Decision Letter 3]

24 Feb 2025

February 21, 2025

Manuscript ID: [PONE-D-24-19264R3] - [EMID:f65fe664dcb5c3e7]

Title: “Exploring Breast Cancer Awareness and Screening Practices Amongst Rural Women in The Gambia: Community-based Cross-sectional Study”

RE: Revised manuscript submission and response to reviewers’ comments

Dear Editor,

This letter is in reference to your email dated February 17, 2025 with reviewers’ comments. We are very pleased that the manuscript is potentially acceptable for publication in PLOS ONE once we have carried out the revisions.

We would like to thank the reviewers for these insightful and helpful comments and for giving us the chance to revise our manuscript. We believe the revised manuscript has been significantly improved and the reviewers’ comments have been addressed adequately. We think in its current form it will make a valuable contribution to the literature on this increasingly important topic.

Please find for your kind consideration the following:

• A section-by-section response to the comments and suggestions of the reviewers (below).

• The revised manuscript, provided as a marked-up copy and a clean copy.

We hope that these changes meet with your favorable consideration. Please do not hesitate to get in touch if you require any further information.

Bakary Kinteh

Corresponding Author

Review’s comments

Abstract:

• Conclusions:

o Line 43 - 44: Authors are advised to include result(s) suggesting the phrase “primarily due to limited knowledge of the urgent need for accessible and affordable screening services” in the Results section of the abstract

Response: We appreciate your feedback. The abstract has been revised to directly reference the identified barriers from the Results section, including a lack of knowledge (58.7%), service unavailability (13.5%), and financial constraints (13.1%), ensuring consistency between the abstract and the results.

Sampling size determination

o Lines 146-148: Authors should ensure the correctness of the following stated figures!

1. Estimated minimum sample size becomes 1103 (not 1092 as indicated) if adjusted for 10% non-response with 1.5 design effect

2. Besides, let’s assume 1092 is correct! Then, only of 1012 (not 1092) that 97.3% response rate would yield 985 completed the study.

Response: We revised and addressed accordingly. We re-calculated the estimated sample size as designed effect of 662*1.5= 993. We later added 10% of 993 gives us 1092 as the estimated sample size. However, we corrected the response rate of 985/1092*100 = 90.2%. Thank you.

Results:

• Table 1 o For sake of clarity to the potential Plos One readers, my previous suggestion is necessary! ▪ Column 2 to be presented as Total (column percentage), while others remain as they were ▪ By so doing, statements on women’s characteristic in lines 240-247 could be referenced adequately.

Response: We revised and included a column percentage of each variable to ensure clarity to PLOS One readers. Thank you.

---

## [Decision Letter · Decision Letter 4]

28 Feb 2025

Exploring Breast Cancer Awareness and Screening Practices Amongst Rural Women in The Gambia: Community-based Cross-sectional Study

PONE-D-24-19264R4

Dear Dr. Kinteh,

We’re pleased to inform you that your manuscript has been judged scientifically suitable for publication and will be formally accepted for publication once it meets all outstanding technical requirements.

Kind regards,

Denekew Bitew Belay, Ph.D

Academic Editor

PLOS ONE

Additional Editor Comments (optional):

Reviewers' comments:

Reviewer's Responses to Questions

**Comments to the Author**

1. If the authors have adequately addressed your comments raised in a previous round of review and you feel that this manuscript is now acceptable for publication, you may indicate that here to bypass the “Comments to the Author” section, enter your conflict of interest statement in the “Confidential to Editor” section, and submit your "Accept" recommendation.

Reviewer #4: (No Response)

2. Is the manuscript technically sound, and do the data support the conclusions?

Reviewer #4: (No Response)

3. Has the statistical analysis been performed appropriately and rigorously? 

Reviewer #4: (No Response)

4. Have the authors made all data underlying the findings in their manuscript fully available?

Reviewer #4: (No Response)

5. Is the manuscript presented in an intelligible fashion and written in standard English?

Reviewer #4: (No Response)

6. Review Comments to the Author

Reviewer #4: The authors are commended for their efforts in improving this manuscript. The revisions are satisfactory.

7. PLOS authors have the option to publish the peer review history of their article (what does this mean? ). If published, this will include your full peer review and any attached files.

**Do you want your identity to be public for this peer review?** For information about this choice, including consent withdrawal, please see our Privacy Policy .

Reviewer #4: **Yes: ** Rotimi Felix Afolabi

---

## [Editor Report · Acceptance letter]

PONE-D-24-19264R4

PLOS ONE

Dear Dr. Kinteh,

I'm pleased to inform you that your manuscript has been deemed suitable for publication in PLOS ONE. Congratulations! Your manuscript is now being handed over to our production team.

Kind regards,

on behalf of

Dr. Denekew Bitew Belay

Academic Editor

PLOS ONE